# On the Effect of Aqueous Ammonia Soaking Pre-Treatment on Continuous Anaerobic Digestion of Digested Swine Manure Fibers

**DOI:** 10.3390/molecules24132469

**Published:** 2019-07-05

**Authors:** Chrysoula Mirtsou-Xanthopoulou, Ioannis V. Skiadas, Hariklia N. Gavala

**Affiliations:** 1Aalborg University Copenhagen, Department of Chemistry and Bioscience, A C Meyers Vænge 15, DK 2450 Copenhagen SV, Denmark; 2Technical University of Denmark, Department of Chemical and Biochemical Engineering, Søltofts Plads 229, 2800 Kgs. Lyngby, Denmark

**Keywords:** ADM1, anaerobic digestion, aqueous ammonia soaking pre-treatment, continuous, digested manure fibers, modelling

## Abstract

(1) Background: The continuously increasing demand for renewable energy sources renders anaerobic digestion as one of the most promising technologies for renewable energy production. Due to the animal production intensification, manure is being used as the primary feedstock for most biogas plants. Their economical profitable operation, however, relies on increasing the methane yield from the solid fraction of manure, which is not so easily degradable. The solid fraction after anaerobic digestion, the so-called digested fibers, consists mainly of hardly biodegradable material and comes at a lower mass per unit volume of manure compared to the solid fraction before anaerobic digestion. Therefore, investigation on how to increase the biodegradability of digested fibers is very relevant. So far, Aqueous Ammonia Soaking (AAS), has been successfully applied on digested fibers separated from the effluent of a manure-fed, full-scale anaerobic digester to enhance their methane productivity in batch experiments. (2) Methods: In the present study, continuous experiments at a mesophilic (38 °C) CSTR-type anaerobic digester fed with swine manure first and a mixture of manure with AAS-treated digested fibers in the sequel, were performed. Anaerobic Digestion Model 1 (ADM1) previously fitted on manure fed digester was used in order to assess the effect of the addition of AAS-pre-treated digested manure fibers on the kinetics of anaerobic digestion process. (3) Results and Conclusions: The methane yield of AAS-treated digested fibers under continuous operation was 49–68% higher than that calculated in batch experiments in the past. It was found that AAS treatment had a profound effect mainly on the disintegration/hydrolysis rate of particulate carbohydrates. Comparison of the data obtained in the present study with the data obtained with AAS-pre-treated raw manure fibers in the past revealed that hydrolysis kinetics after AAS pre-treatment were similar for both types of biomasses.

## 1. Introduction

Anaerobic digestion is one of the most promising renewable energy technologies, as it provides a solution to both environmental and energy considerations. Some of the benefits of this technology are that it reduces the odour, minimizes the size of the organic wastes, contributes to the reduction of the greenhouse gases emissions and produces a high value fertilizer as well as a renewable energy gas. Anaerobic digestion of swine manure has been a wide matter of discussion and it has proved a promising approach for renewable energy production in the form of methane. Nowadays, due to the intensification of animal production, manure seems to be the largest available substrate for biogas plants, in Europe. According to the European parliament [1] 55% of the available biomass for biogas production comes exclusively from animal manure. In fact, 22 large-scale biogas plants are currently under operation in Denmark using manure as primary feedstock, but their economical profitable operation relies on increasing the methane yield from manure [2].

Although manure is undoubtedly an excellent carrier for biogas plants, it contains an organic fraction which is not easily degraded. Many different strategies have been developed so far to increase the methane potential of the manure. New separation technologies that are being applied before anaerobic digestion have been tested over during the last few years [3]. The liquid fraction of the manure could be used as fertilizer in the farms while the solid fraction could be transported to centralized biogas plants for methane production [4]. Additionally, several different pre-treatment technologies have been applied to increase the methane yield of manure and manure fibers. Some representative results are those of Hartmann et al., [5], where mechanical maceration was applied on fibers with a 25% increase on the methane yield, and also the study of Bruni et al., [6] who applied hydrothermal, chemical, as well as enzymatic pretreatments on fibers, with a maximum increase of 66%. One chemical pre-treatment, which has been initially investigated for bioethanol production is Aqueous Ammonia Soaking (AAS). Ammonia is a proven delignification reagent, which is quite safe to handle, non-polluting and non-corrosive [7], and due to its high volatility, it can be easily recovered and recycled [7], thus avoiding the need of any further chemical consumption. More recently, AAS was also applied on the treatment of the fibrous fraction of raw and digested manure fibers, under room temperature [8,9]. In both studies, Aqueous Ammonia Soaking (AAS) treatment combined with subsequent removal of ammonia (which also gives the possibility for being recycled in a full-scale plant) was applied effectively, resulting in a high increase of the methane yield. According to Jurado et al. [8] an increase of up to 78% was observed compared to the non-treated raw manure fibers (control), during the first 16 days of digestion while optimisation of the pre-treatment conditions for raw manure fibers resulted in an impressive 244% increase of the methane yield [10]. Similar results were observed in the study of Mirtsou-Xanthopoulou et al. [9] where an increase of up to 110% was observed in the methane yield of digested manure fibers compared to the non-treated ones (control). Post-treatment of digested fibers focuses only on hardly biodegradable biomass in contrast to raw fibers where easily biodegradable material is also present. Therefore, the mass of fibers to be treated is expected to be significantly lower in case of digested fibers leading thus to a more economical process. The question that the present study addresses is whether models developed for digesters processing manure can actually be applied when the influent stream is supplemented with AAS-treated digested fibers and how disintegration/hydrolysis kinetics compare to those of AAS-treated raw fibers.

In general, mathematical models are used in order to efficiently describe and predict the efficiency of different biological processess. Different models for describing the anaerobic digestion process and predicting the methane production and effuent characteristics started to emerge as early as in 1970 [11]. The Anaerobic Digestion Model 1 (ADM1) is a generic mathematical model, which was developed specifically for the anaerobic digestion process in 2002 by the IWA anaerobic digestion task group [12]. ADM1 is a modeling platform that can be used for simulating the biogas process under different substrates and operating conditions, taking into account the most important biological and physicochemical processes of anaerobic digestion. Since ADM1’s publication, numerous studies have been published applying, fitting and tuning ADM1 to different feedstocks, reactors and operating conditions [13]. Digesting manure with different substrates is a case that has also been lately applied in modeling with ADM1 [14,15,16,17,18,19], although to a lesser extent considering that there are around 300 ADM1 based studies and only 10% refer to manure as the main feedstock despite the fact that manure is one of the basic and most well-known feedstocks of anaerobic digesters.

In the present study, continuous experiments at a mesophilic (38 °C), CSTR-type, anaerobic digester fed with swine manure first and a mixture of manure with AAS-treated digested fibers in the sequel, were performed in order to verify the experimental data obtained from batch experiments. A modified Anaerobic Digestion Model 1, accounting for co-digesting feedstocks of different composition [20] was used in order to assess the effect of the addition of AAS-pre-treated digested manure fibers on the kinetics of anaerobic digestion process.

## 2. Results and Discussion

### 2.1. Influent Characterization

The characteristics of manure and AAS-treated digested manure fibers are shown in Table 1. As anticipated, the AAS-treated fibers consisted mainly of particulate organic matter with carbohydrates being a substantial fraction while manure was rich in soluble organic matter.

### 2.2. Continuous Experiments and Modeling Results

Operating conditions and steady state characteristics of the anaerobic digester fed with manure A (first phase) and a mixture of manure A and AAS-pretreated digested manure fibers (second phase) are summarised in Table 2. Despite the fact that manure fibers were characterised by a much lower content in soluble organic material (Table 1) the biogas production of the second steady state (with manure and fibers as influent) was 16% higher than that of the first (with only manure as influent). The methane yield was calculated as 0.303l CH_4_/g TS during the first steady state with manure as influent and 0.272 l CH_4_/g TS during the second steady state when the digester was being fed with mixture of manure and AAS-treated digested fibers. Given that the TS ratio of manure:fibers in the influent of the digester was 0.52:0:48 and assuming that the methane yield due to the manure fraction was 0.303l CH_4_/g TS also during the second steady state, then the methane yield of AAS-treated digested fibers was calculated to be 0.238 l CH_4_/g TS. In previous studies, the methane yield from AAS-pretreated digested fibers was measured in the range of 141–160 mL CH_4_/g TS in batch experiments (compared to a methane yield of 76 mL CH_4_/g TS from non-treated digested fibers) [9]. Consequently, the continuous experiments verified what it was observed in batch experiments; namely that AAS pretreatment resulted to a significant increase of methane yield of digested fibers. The even higher methane yield obtained from the continuous experiment was most probably due to the adaptation of the mixed microbial culture on the different feedstock since the batch experiments were performed with inoculum which was adapted to manure only, where actually no hydrolysis had taken place [20].

ADM1 as developed in Jurado et al. [20] was used to simulate the first phase where only manure A was being added in the reactor. During the second phase, where the digester was fed with the mixture of manure A and AAS-pretreated digested fibers, fitting of the ADM1 to the experimental data was performed. The only kinetic parameters, which were mostly affected, were those of carbohydrates and proteins hydrolysis. Hydrolytic kinetic constants for carbohydrates and proteins in manure had been calculated as 0 and 2.8 × 10^−4^ d^−1^, respectively [20], while for the mixture were increased to 6.8 × 10^−2^ and 7.0 × 10^−3^ d^−1^ and are in very good agreement with the hydrolysis constants estimated for AAS-pretreated raw manure fibers, 7.3 × 10^−2^ and 7.1 × 10^−3^ d^−1^, respectively [20]. Hydrolysis constants were remarkably lower compared to all soluble substrate consumption rates, including volatile fatty acids and therefore it can be assumed that hydrolysis is the rate-limiting step for the overall anaerobic digestion process when adding manure fibers at the ratio tested hereby. Kinetics of volatile fatty acids uptake seemed to be unaffected as the behaviour of the digester under the pulse disturbances was satisfactorily simulated by the model without any further change in the kinetic parameters. The latter is very well depicted in Figure 1, Figure 2 and Figure 3 where the experimental and predicted by the model VFA concentrations during the pulse experiments are presented.

Model predictions for the steady state on manure A are also shown in Table 2. It is noticeable that the model, which was developed in another digester fed with manure, was able to accurately predict the steady state reached in the present study. Table 2 also includes model predictions for the second phase, after fitting of the model had been performed. It is obvious that the model was able to adequately predict the steady state reached on the mixture of manure A and AAS-pretreated digested fibers after alteration of just the hydrolytic constants.

Subsequently, ADM1, with the new values of the hydrolytic constants as obtained from the second phase, was used to simulate the performance of the digester during the third and fourth phases, where a mixture of manure B and AAS-treated digested fibers (at a ratio of 0.52:0.48 on Total Solids basis) was the influent at an HRT of 25 and 11.2 d., respectively.

The experimentally measured, as well as the calculated by the model, biogas production rate together with the organic (TS based) loading rate of the digester throughout the four phases of the continuous experiment are shown in Figure 4. The model exhibited a remarkable ability to predict the biogas production rate after significant change of the feeding characteristics (from manure A to manure B in the third phase) as well as of the HRT of the system (from 25 d to 11.2 d in the fourth phase). Last but not least, the model also satisfactorily predicted the methane concentration in the biogas, of the last two phases of the continuous experiments. Specifically, for the third phase, the model predicted an average percentage of methane in the biogas of around 64.6 ± 0.5 and experimentally it was measured as 63.2 ± 0.1. For the fourth phase, the methane percentages predicted from the model and measured experimentally were 68.2 ± 2.2 and 63.2 ± 0.4, respectively.

## 3. Materials and Methods 

### 3.1. Feedstock

Digested manure fibers (fibers collected after anaerobic digestion) and two batches of swine manure (A and B) were kindly provided by Morsø BioEnergi and stored at −20 °C until used. The two batches of manure were collected at a different period of the year, therefore their characteristics varied considerably—as shown in Table 1.

### 3.2. Analytical Methods

Characterization of the liquid fraction of manure and manure fibers included the determination of the soluble components and was done after centrifugation of the samples at 10,000 rpm for 10 min and filtration of the supernatant through 0.2 μm membrane filters.

Total (TS) and volatile (VS) solids were measured according to standard methods [21]. Total and soluble Chemical Oxygen Demand (COD) was determined using Hach Lange kits LCK_914 (5−60 g L^−1^ range) and LCK_514 (100−2000 mg L^−1^ range), respectively.

For total phosphorus and total nitrogen determination the material was dried at 42 °C overnight and powdered, while for the soluble and inorganic measurements, the material was centrifuged at 10,000 rpm for 10 min and the supernatant was passed through 0.2 μm membrane filters. For the determination of the non-soluble and soluble organic phosphorus persulphate digestion was applied to the solid and liquid fraction, respectively, followed by ascorbic acid photometric determination of phosphate ions [21]. For the determination of non-soluble and soluble organic nitrogen, digestion with the micro-kjeldahl apparatus was applied to the solid and liquid fraction, respectively, followed by distillation and titration (titrimetric method) for ammonium ions determination (APHA, 2005). For the inorganic forms of phosphorus (PO_4_^−3^-P) and nitrogen (NH_4_-N) determination, analysis was carried out by applying ascorbic acid photometric determination [21] and by using the Hach Lange kit LCK_305 (1–12 mg L^−1^ range), respectively.

Two groups of carbohydrates were determined in the samples of manure and manure fibers: the first group was the total carbohydrates, including those bound in the lignocellulosic biomass and the second group was the simple sugars [22]. Analysis of the two groups of carbohydrates was carried out based on the NREL analytical procedures [23]. Detection and quantification of sugar monomers (glucose, xylose and arabinose) was made with HPLC-RI equipped with an Aminex HPX-87H column (BioRad, Hercules, CA, USA) at 60 °C. A solution of 4 mmol L^−1^ H_2_SO_4_ was used as eluent at a flow rate of 0.6 mL min^−1^. Samples for HPLC analysis were acidified with a 10% *w/w* solution of H_2_SO_4_, centrifuged at 10,000 rpm for 10 min and finally filtered through a 0.45 μm membrane filter. Klason lignin was also determined according to NREL analytical procedures [23]. For the determination of both total carbohydrates and lignin the material was dried at 42 °C overnight and powdered, while for the determination of free sugars the material was centrifuged at 1000 rpm and the supernatant was passed through 0.2 μm size membrane filters.

For the quantification of Volatile Fatty Acids, VFA, 1 mL of sample was acidified with 17% H_3_PO_4_ and filtered through minisart high flow filter (pore size 0.5 µm). VFAs were analyzed on a gas chromatograph (PerkinElmer Clarus 400) with a flame ionization detector and a capillary column (Agilent HP-FFAP column, 30 m long and 0.53 mm inner diameter).

Biogas composition in methane was measured with a gas chromatograph (SRI GC model 310) equipped with a thermal conductivity detector and a packed column (Porapak-Q, length 6 ft and inner diameter 2.1 mm). The temperature for injector, column and detector was set to 80 °C. The volume of methane produced in sealed vials during methane potential tests was calculated by multiplying the biogas composition in methane with the headspace volume.

### 3.3. Experimental Set-Up

#### 3.3.1. Ammonia Pre-Treatment

Samples of digested manure fibers were soaked in ammonia reagent (32% *w*/*w* in ammonia) at a ratio of 10 mL reagent per g TS, under room temperature. The treatment was performed in closed glass flasks to avoid ammonia evaporation. After the completion of the treatment, water was added at a ratio of 10 mL per g TS to facilitate the subsequent ammonia distillation step. Distillation was performed using a rotary evaporator (Buchi RII Rotavapor) with a vertical condenser.

#### 3.3.2. Continuous Experiments

One mesophilic (38 °C) CSTR-type digester of 3 l useful volume was started-up using mixed anaerobic liquor from a swine manure treating digester and fed with swine manure A at a hydraulic retention time of approximately 25 d (first phase). After the digester reached steady-state, the influent was changed (second phase) to a mixture of swine manure A and AAS-treated digested manure fibers (at a ratio of 0.52:0.48 on Total Solids basis). The ratio has been chosen based on the maximum addition of fibers allowing for a smooth operation (no blockings of the flow of material in the pipes) of the lab-scale digesters and pumps. Subsequently and after the digester reached the second steady state, the manure provided by Morsø BioEnergi changed to a new batch of manure (B) with much lower TS content (1.63 gTS/100 g manure) than the one used for the first and second phase (4.4 gTS/100 g manure). The digester was allowed to reach steady state (third phase) with the new type of manure under the same operating parameters applied for the second phase (TS ratio of manure: AAS treated fibers and HRT). After the third steady state was reached, the HRT was reduced to 11.2 d (fourth phase).

Summarizing, the four phases involved are as following:First Phase: manure A as influent at an HRT of 25 d.Second Phase: mixture of manure A and AAS-treated digested fibers (at a ratio of 0.52:0.48 on Total Solids basis) as influent at an HRT of 25 d.Third Phase: mixture of manure B and AAS-treated digested fibers (at a ratio of 0.52:0.48 on Total Solids basis) as influent at an HRT of 25 d.Fourth Phase: mixture of manure B and AAS-treated digested fibers (at a ratio of 0.52:0.48 on Total Solids basis) as influent at an HRT of 11.2 d.

The feeding of the digester was intermittent and repeated once a day during all phases. The operation, however, can actually be considered as “continuous” due to the relatively long retention times applied. Daily monitoring of the digester included biogas production and composition in methane, pH, volatile fatty acids and soluble COD concentration. When the digester reached the second steady state while fed with the mixture of manure and fibers, complete characterisation was performed in terms of almost all measurable components of ADM1 (see paragraph 3.4.2). Subsequently, the digester was subjected to pulse disturbances of acetic, propionic and butyric acids and soluble influent fraction (obtained after centrifugation and filtration) in order to study the dynamics of the processes (those experiments are considered part of the second phase). Following this, 50 mL concentrated solutions of acetic, propionic and butyric acids and a 600 mL sample consisting of the soluble fraction of the mixture (manure and digested fibers), were consecutively injected into the digester in order to increase the reactor concentration of the injected substrate to around 1600, 1200 and 1600 mg COD/l for the acetic, propionic and butyric acids, respectively, and to around 400 mg COD/l for the soluble fraction of the mixture [24]. The response of the bioreactor to these disturbances was monitored mainly through measurements of biogas production and composition in methane, pH and VFA concentration until all components approach their levels before the pulse.

### 3.4. Modeling

Anaerobic Digestion Model No1 (ADM1) was used to simulate the anaerobic digestion process and to assess the effect that the addition of AAS pretreated digested fibers had on the kinetic parameters. In the study by Jurado et al. [20], IWA anaerobic digestion model (ADM1) had been fitted to the experimental data obtained from a swine manure-fed digester which was different than the one used in this study and hydrolysis constants of carbohydrates (khydr_ch), proteins (khydr_pr) and lipids (khydr_li) and maximum uptake rates of long chain fatty acids (km_fa) and volatile fatty acids (km_c4, km_pro and km_ac for butyric, propionic and acetic acid, respectively) had been calculated as shown in Table 3. That model was used to simulate the first steady state of the reactor (with only swine manure as influent). Subsequently, the model was fitted to the experimental data obtained throughout the second phase, where a mixture of manure and fibers was introduced as influent and new hydrolysis constants had been obtained. Finally, the experimental results obtained when the new batch of manure (B) was added as well as when the Hydraulic Retention time (HRT) was reduced to 11.2d (third and fourth phase, respectively) were used to validate the model fitted to the second phase. The software used was Aquasim 2.1 g and the secant method was applied for the parameters estimation.

#### 3.4.1. Correction of the HRT in the Kinetic Model

Due to the structural characteristics of the digester, there was an accumulation of solids, which was quantified by measuring Total Suspended Solids (TSS) concentration in the interior (by taking sample directly from inside the digester) and the effluent of the reactor. The higher retention time of the solids was expressed by introducing a new variable into the model, called ‘tres’. Tres was calculated by substracting the Hydraulic Retention Time (HRT) from the Solids Retention Time (SRT) (Equation (1)): *tres* = SRT − HRT(1)

In the model, it was considered that solids and microbial biomass were recycled in the reactor and their concentration was corrected according to Equation (2) [25].
(2)$$x=x\cdot \left(Q_{out}-\frac{V_{reactor}}{tres+\frac{V_{reactor}}{Q_{out}}}\right)$$
where *x* is the solids and biomass recirculation flow (kg COD/d), *V_reactor_* is the reactor active volume (m^3^) and *Q_out_* the effluent flow (m^3^/d).

#### 3.4.2. Organic and Inorganic Inputs in the Kinetic Model

Swine manure characteristics were considered as measured and calculated in Jurado et al. [20]. AAS-treated digested manure fibers were characterized in terms of total and soluble COD, total carbohydrates and free sugars, total and soluble Kjendahl nitrogen and NH_3_-N, volatile fatty acids (valeric, butyric, propionic and acetic acids), inorganic phosphorus and inorganic carbon. Particulate and soluble inerts were determined as the residual COD after three months of batch anaerobic digestion. Particulate carbohydrates and proteins were calculated as the difference between total and free carbohydrates/sugars and total and soluble Kjeldahl nitrogen, respectively. Aminoacids were calculated as the difference between soluble Kjeldahl nitrogen and NH_3_-N. Calculation of particulate lipids was based on the difference between non-soluble COD (total − soluble COD) and the sum of particulate carbohydrates, proteins and inerts, while long chain fatty acids were calculated as the difference between soluble COD and all measured soluble components (sugars, aminoacids, volatile fatty acids and soluble inerts).

In the model, the influent concentration of all organic and inorganic constituents was initially expressed in kg COD/100 kg TS and kmole/100 kg TS, respectively, and subsequently was multiplied with the influent TS concentration and calculated in kg COD/m^3^ and kmole/m^3^, respectively. This was done in order to take into account the variations of the TS concentration in the feeding. The influent flow rate, Q_in_, was given in m^3^/d and the flow rate of each individual component (in kg COD/d and kmole/d for organic and inorganic substances, respectively) was calculated as the product of the volumetric concentration in the influent with the influent flow rate, Q_in_.

## 4. Conclusions

Continuous experiments at a mesophilic (38 °C) CSTR-type anaerobic digester fed with swine manure first and a mixture of manure with AAS-treated digested fibers in the sequel, were performed. The methane yield of AAS-treated digested fibers under continuous operation was 49–68% higher than that calculated in batch experiments in the past. Anaerobic Digestion Model 1 (ADM1) previously fitted on manure fed digester was used and simulated very satisfactorily the first experimental phase, proving thus the robustness of the model and process. Fitting of the second experimental phase, where the influent stream was supplemented with AAS-treated digested fibers (digested fibers refer to manure fibers after anaerobic digestion), indicated that AAS treatment had a profound effect mainly on the hydrolysis rate of particulate carbohydrates and it is noticeable that the estimated disintegration/hydrolysis constants were comparable to those obtained with AAS-treated raw manure fibers (raw fibers refer to manure fibers before anaerobic digestion). Last but not least, pulse experiments with important intermediates of the process, i.e. acetic, propionic and butyric acids, indicated that the associated kinetic constants were not affected by the addition of the AAS-treated digested fibers in the influent.

## Figures and Tables

**Figure 1 molecules-24-02469-f001:**
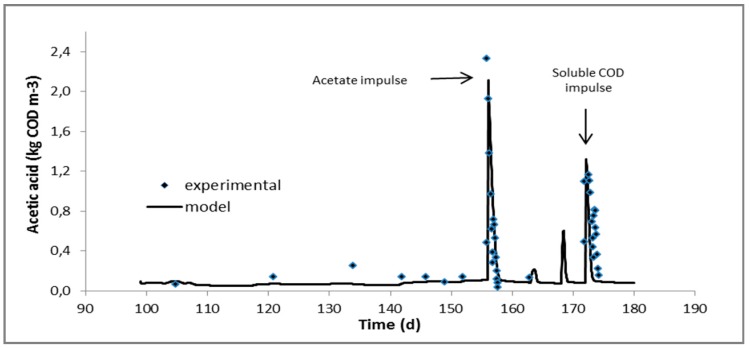
Experimental and predicted by the model acetic acid concentration during the experiment with mixture of manure and AAS-treated digested fibers as influent.

**Figure 2 molecules-24-02469-f002:**
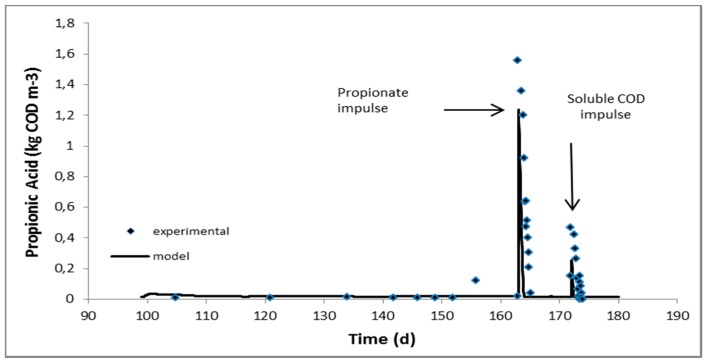
Experimental and predicted by the model propionic acid concentration during the experiment with mixture of manure and AAS-treated digested fibers as influent.

**Figure 3 molecules-24-02469-f003:**
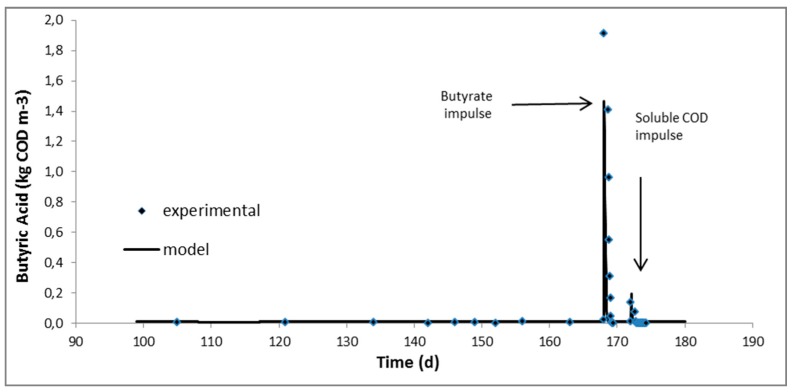
Experimental and predicted by the model butyric acid concentration during the experiment with mixture of manure and AAS-treated digested fibers as influent.

**Figure 4 molecules-24-02469-f004:**
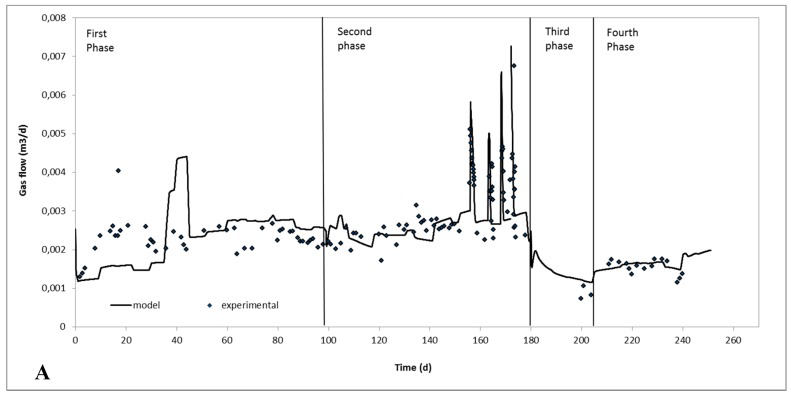
Experimental and theoretical biogas production rate in m^3^/d (**A**) and organic loading rate in g TS/d (**B**) throughout the four phases.

**Table 1 molecules-24-02469-t001:** Characteristics of manure (A) and (B) and of Aqueous Ammonia Soaking (AAS)-pre-treated raw manure fibers.

Characteristics, Particulate Matter	Manure A ^a^ (Used in Phases 1 and 2)	AAS-Pretreated Digested Fibers (Used in Phases 2, 3 and 4)	Manure B ^a^ (Used in Phases 3 and 4)
COD, kg/100 kgTS	79.11	75.34	124.19
Carbohydrates, kg COD/100 kg TS	6.10	32.75	9.57
Proteins, kg COD/100 kg TS	28.00	21.41	43.95
Lipids, kg COD/100 kg TS	32.65	1.10	51.25
Inerts, kg COD/100 kg TS	12.37	20.08	19.42
Characteristics, soluble matter			
COD, kg/100 kgTS	91.83	23.7	55.90
Sugars, kg COD/100 kg TS	0	0	0.0
Aminoacids, kg COD/100 kg TS	18.92	1.02	11.52
Long chain fatty acids, kg COD/100 kg TS	21.25	19.49	12.94
Valeric acid, kg COD/100 kg TS	0	0.66	0.0
Butyric acid, kg COD/100 kg TS	3.98	0.00	2.42
Propionic acid, kg COD/100 kg TS	8.46	0.34	5.15
Acetic acid, kg COD/100 kg TS	34.75	2.07	21.15
Inerts, kg COD/100 kg TS	4.46	0	2.71
Inorganic carbon, kmole/100 kg TS	0.53	0.05	0.32
Inorganic phosphorus, kmole/100 kg TS	9.59 × 10^−3^	6.52 × 10^−3^	0.0058
Inorganic nitrogen (NH_3_-N), kmole/100 kg TS	0.73	4.28 × 10^−2^	0.44

^a^ as reported in Jurado et al. [20].

**Table 2 molecules-24-02469-t002:** Operating conditions, experimental measurements and model predictions for the steady states on manure and mixture of manure and AAS-treated digested fibers.

Operating Conditions	Manure A (1^st^ Phase)		Mixture of Manure A and AAS-Treated Digested Fibers (2^nd^ Phase)	
HRT, d	24–25		24–25	
Flow rate, mL/d	114–134		117–125	
Organic loading rate (OLR), g COD/l/d	2.83–3.53		2.30–3.65	
Soluble OLR, g soluble COD/l/d	1.52–1.90		1.00–1.58	
Steady state characteristics	Manure A (1^st^ phase)		Mixture of manure and AAS-treated digested fibers (2^nd^ phase )	
	Experimental	Model	Experimental	Model
Methane, %	67.6 ± 2.1	64.2 ± 8.3	66.6 ± 1.8	67.0 ± 7.6
Biogas production, m^3^/d	2.32 × 10^−3^ ± 0.2 × 10^−3^	2.65 × 10^−3^ ± 0.1 × 10^−3^	2.70 × 10^−3^ ± 0.3 × 10^−3^	2.9 × 10^−3^ ± 0.2 × 10^−3^
pH	8.2	8.1	8.1	8.0
Acetic acid, kgCOD/m^3^	0.142 ± 0.025	0.087 ± 0.013	0.161 ± 0.084	0.081 ± 0.018
Propionic acid, kgCOD/m^3^	0.023 ± 0.005	0.018 ± 0.002	0.012 ± 0.003	0.019 ± 0.002
Butyric acid, kgCOD/m^3^	0.026 ± 0.002	0.011 ± 0.001	0.007 ± 0.0001	0.012 ± 0.001

**Table 3 molecules-24-02469-t003:** Kinetic parameters as calculated in the study of Jurado et al. for a manure fed anaerobic digester [20].

Kinetic Parameter	Units	Value
Carbohydrates hydrolysis constant, khydr_ch	d^−1^	<<
Proteins hydrolysis constant, khydr_pr	d^−1^	3.0 × 10^−3^
Lipids hydrolysis constant, khydr_li	d^−1^	2.8 × 10^−4^
Maximum uptake rate of long chain fatty acids, km_fa	kg COD fa/kg COD x_fa/d	0.93
Maximum uptake rate of butyric acid, km_c4	kg COD c4/kg COD x_c4/d	13.1
Maximum uptake rate of propionic acid, km_pro	kg COD pro/kg COD x_pro/d	6.56
Maximum uptake rate of acetic acid, km_ac	kg COD ac/kg COD x_ac/d	45.02

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
