# Peer review of "On the Effect of Aqueous Ammonia Soaking Pre-Treatment on Continuous Anaerobic Digestion of Digested Swine Manure Fibers"

_molecules, 2019, doi:10.3390/molecules24132469_

Reviewer 1 Report

In this article, anaerobic digestion of manure and its mixture with residual fibers treated with aqueous ammonia (AAS) was studied. Besides, ADM1- calculation model was tested. The conclusions were the following:

-          After treatment of residual fibers using AAS-method, their digestibility   increases.

-          The methane yield at continuous operation from AAS-treated fibers was higher than that calculated in batch experiments.

-          It was shown that calculation of digestion process by means of ADM1-model gives results very close to experiments.

Nevertheless, consideration of this article raises some questions.

(1). Why ratio between manure and AAS-treated fibers was chosen 0.52:0.48 and not fifty-fifty or some other?  

(2). TS content in manure substrates was 4.4 g TS/ 100g for manure A and 1.63 g TS/100g for manure B. This TS content of manure samples was used by chance or special?

(3). In Table 2 was shown that manure A has 79.11 COD kg/100 kg TS, while Manure B 124.19 COD kg/100 kg TS. Has a manure sample been prepared specially by adding a higher content of proteins, lipids and other substances? Or this sample was purchased with so high COD content? Why for next digestion stages just substrate with higher COD should be needed? The explanation is needed.

Regarding soluble mater (Table 2). The method of determination of such matter should be described including temperature and time of dissolution.

(4). If compare Tables 3 and 4, the difference in experimental methane yield can be seen. For manure A - 67% (Table 3) and 67.6% (Table 4); and for manure B - 65% (Table 3) and 66.6% (Table 4). Why there is such a difference?

Decision: This paper can be published after minor revision.

Reviewer 2 Report

The paper entitled “On the effect of Aqueous Ammonia Soaking 2 pretreatment on continuous anaerobic digestion of 3 digested swine manure fibers” studied the valorization of swine manure through the anaerobic digestion process to obtain methane rich-biogas. Anaerobic digesters have been fed with fresh swine manure individually and in co-digestion with pretreated digested fibers, using aqueous ammonia soaking pretreatment. The study was carried out at a mesophilic temperature and a semi-continuous mode of operation. Moreover, a kinetic study was applied by using an ADM1 model in order to study the effect of supplementing the digester fed with swine manure with the digested fibers previously pretreated with aqueous ammonia soaking.

The study is very interesting and the paper is well structured and written, providing enough data and suitable discussion. I think that some aspects could be improved by authors before publication. My specific comments are listed below.

Abstract

Information about which substrates have been used and if assays were in batch or in continuous is not clearly addressed in the abstract, which make the understanding difficult. I suggest modifying and clarifying this aspect in order to make it clear for the reader from the beginning. In this study the feeding strategy was once a day, so it is a semi-continuous assay and not continuous.

P1:L17-L18: Please remove this sentence “full-scale anaerobic digester to enhance their methane productivity in batch experiments.”, if not, you should provide a suitable reference and this is not usual in abstracts.

Why authors have used “digested fibers separated from the effluent of a manure-fed” as substrate?

Introduction

P2:L47-L48: I am not sure that the liquid fraction of manure could be directly used as fertilizer. There are some norms which should be considered and accomplished before using any substrates/material as fertilizer.

P2:L56: A missing word after due “to”

Materials and methods

This section is very well detailed.

P5:L95-L96: What do you mean by “in the interior and the effluent of the reactor”? Please clarify.

P5: L206 and 210: Please, correct the “Kjendahl nitrogen”, it is “Kjeldahl nitrogen”

Results and discussion

P6:L231: “Figure 1 shows the volumetric and the organic (TS based) loading rate”. Why showing volumetric loading rate? I suggest representing the evolution of biogas/methane together with HRTs or OLRs.

Tables 3 and 4: The percentages of methane do not match in both tables, there is a small difference and data should be exact. Moreover, why repeating the same data in the two tables? Biogas parameters and pH are repeated. I suggest revising and improve the first part of the results and discussion. I also suggest separating experimental results and modeling in two subsections.

Conclusions

The conclusion is very long. I suggest resuming by showing only relevant findings.
